# Production of Large-Sized Ceramic Stones Based on Screenings from Waste Heap Processing Using the Technology of Stiff Extrusion for Molding Products

Khungianos Yavruyan * and Evgeniy Gaishun

Faculty "Engineering and Construction", Don State Technical University, 344003 Rostov-on-Don, Russia
* Correspondence: khungianos@mail.ru

**Abstract:** This article discusses the prospects for using large-format ceramic stones in the construction of contemporary homes, as well as an overview of raw materials and technologies for production. The most promising technology is stiff extrusion with the ability to load the raw products on firing trolleys and accelerate the processes of firing and drying. Characteristics of Eastern Donbass waste heaps processing screenings are given, which are coal mining by-products and are considered to be main raw material for the production of large-sized ceramic stone. It is shown that as a result of introduction of waste heaps into the production of ceramic stones with the lowest prime cost, the density of the resulting products will be less than 800 kg/m$^3$, the thermal conductivity will be less than 0.20 m·°C/W, and the strength grade will be M150 and higher. Thus, the use of ceramic stones in total volume of wall products for residential construction will reach the level of 80% and will increase the competitiveness of the material, especially when compared with gas silicate products, as used in Western Europe. High economic feasibility of the production of such materials based on by-products of waste heap processing is shown as well.

**Keywords:** ceramic block; screening; siltstone; coal; coal sludge; stiff technology; density; strength; water absorption





## 1. Introduction

At the beginning of this century, there was an intensive construction of new enterprises engaged in the manufacture of large-sized highly porous ceramic stones in the industry of wall ceramic materials. In Russian civil engineering, ceramic bricks with a lower density and thermal conductivity replaced concrete wall structures, but in Western Europe this happened in the 1980s and 1990s. An objective comparison of ceramic blocks with analogues in the use of structures and all stages of construction will justify this trend. Ceramic stones account for about 80% of the total volume of wall products used for housing construction in Western Europe. As for the Russian construction industry, this figure is significantly lower. The main factor in this case is the cost, which, when compared with gas silicate products, is slightly higher [1,2]. This is often the primary reason for builders, when evaluating only the construction itself without considering the long-term operation of the building [3–8].

In recent years, the most popular products in civil engineering appeared to be highly efficient ceramic blocks, which are classified as ceramic stones based on GOST 530-2012. The demand for ceramic blocks is explained by their properties: average density should be less than 800 kg/m$^3$, calculated thermal conductivity—less than 0.20 W/m·°C, high frost resistance, and a fairly high strength grade—M75 and higher, which allows them to be used for connecting load-bearing structures. The advantage of blocks is their size. The most popular are blocks with a length of 380 mm and 510 mm, a width of 250 mm, and thickness of 229 mm. In terms of ordinary bricks, this is 10.7 and 14.3 pieces of products. A wall made of such blocks with a facing-in half a brick fully meets the requirements for thermal performance.

There are no factories for the production of ceramic blocks in the Rostov region, although the minimum requirement for the city of Rostov-on-Don alone is about 50 million pieces in terms of ordinary bricks. The region covers the need for blocks with deliveries from the Krasnodar Territory (Slavyansk-on-Kuban), Volgograd, Kabardino-Balkaria (Prokhladny). An important point for ceramic blocks to successfully compete with gas silicate blocks should be their cost—it should not exceed 3000 rubles per 1 m$^3$. This is in terms of an ordinary brick no more than six rubles apiece. The cost of freight transportation has recently risen significantly, which is why it is necessary to create a place for the manufacture of ceramic stones with lower cost products in the Rostov region. The implementation of this is possible only if the following cost conditions are reduced: technological operations, raw materials, firing, and the construction of new plants or improvement of old plants.

For many years VNIIStrom of P. P. Budnikova, JSC have been engaged in the technology of processing and using technogenic wastes of the coal series with different contents of the combustible component. Based on its mineral content it has been used as either the main raw material or as an additive. It has been established that when the content of the combustible component is 5–6%, coal waste can be used as the main raw material [9,10].

The Spanish company "Agemac" uses coal waste in the production of ceramic facing bricks. Products with increased strength characteristics were obtained (compressive strength 45 MPa). Fuel consumption during firing was reduced by 50% [11].

The French company "Ceric" has developed technology and equipment for the production of stiff molded bricks from coal waste [12].

The purpose of the research is to scientifically substantiate the feasibility of manufacturing reduced-density large-format ceramic blocks from by-products of waste heap (coal dumps) processing using stiff extrusion technology.

The novelty of the work is the scientific substantiation of production of ceramic stones from by-products of waste heap processing using technology of stiff extrusion.

## 2. Materials and Methods

A technological development with a goal of ceramic block production has recently taken place at Don State Technical University. It was conducted based on raw materials made from by-products of waste heap processing in Eastern Donbass, in order to achieve high profitability and minimize production costs.

Due to the economic aspect, screenings of waste heap disposal are considered to be extremely appealing raw material for the manufacture of wall ceramics and other products. They are obtained as a result of processing waste heaps during coal mining. During the processing other materials also appear, which are separated according to grain composition, the amount of coal component, and the mineralogical and petrographic composition [13,14].

Screenings are materials with medium grain size varying from 2–5 mm and are represented in the Eastern Donbass by siltstones mixed with mudstones. They do not contain more than 1% of coal. Therefore, they are the least of all in demand after waste heap processing. Coarse-grained materials with a grain size of more than 5 mm are widely used in industrial civil and road construction, and fine-grained material with a grain size of less than 2 mm serves as low-calorie fuel when screenings are already less in demand. They cost almost nothing, and factories have to spend money to store them. In relation to the prerequisites indicated earlier, we have performed a number of studies on the production of large-sized ceramic stones from screenings with reduced thermal conductivity and density.

Investigation of waste heap processing screenings to establish the possibility of producing ceramic blocks on their basis were carried out according to the following scheme:

- Determination of structural features, chemical and mineralogical composition;
- Study of pre-fired ceramic properties using plastic and compression-molding methods in relation to plastic and low-plastic stone-like raw materials;
- Study of firing ceramic properties;
- Selection of raw materials with testing of laboratory samples;

- Characteristics of finished products, establishing relationships, determining the influence of various technological factors on them;
- Development of primary technological production schemes.

Screenings from waste heap processing in the Eastern Donbass region do not practically differ from clay raw materials in terms of chemical composition (Table 1). They contain almost the same amount of silicon and aluminum oxides as semi-acidic clay raw materials. Iron oxide makes up the same proportion as it does in the composition of each dark-burning clay raw material. Due to the presence of potassium feldspars and feldspars with a predominance of potassium oxide, the presence of alkali oxides is almost the same as in refractory and low-melting clay raw materials.

**Table 1.** The average chemical composition of waste heap processing screenings excluding the coal content.

| Loss on Ignition | SiO$_2$ | Al$_2$O$_3$ | Fe$_2$O$_3$ | CaO | MgO | SO$_3$ | K$_2$O | Na$_2$O |
|---|---|---|---|---|---|---|---|---|
| 3.2–6.3 | 56.5–64.8 | 17.0–22.2 | 3.1–6.3 | 1.1–3.3 | 0.5–2.1 | 0.4–1.0 | 2.1–4.6 | 0.8–2.0 |

Screening minerals are represented by feldspars, quartz, micas and hydromicas. Albite and orthoclase make up the majority of feldspar. Pelletization and chloritization are two significant secondary alterations that they are susceptible to. Secondary ferrous minerals (oxides and hydroxides) are also present. A grain from screening is shown on an electron micrograph (Figure 1), where the sizes of the components can be clearly seen. Figure 2 shows X-ray diagrams of common screenings of waste heap processing from different companies of eastern Donbass.

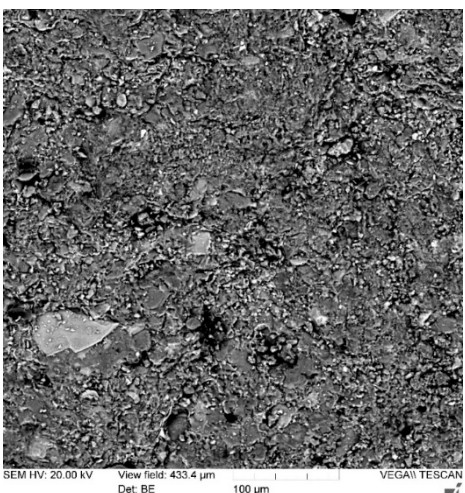

**Figure 1.** Micrograph of a grain from screening from Pyramid LLC (Rostov region, Russia).

Diffraction peaks of around 10 and 5 Å can be used to identify hydromicas (illite) and micas. Peaks of diffraction for feldspar and plagioclase are 3.20, 4.04, and 2.96 Å. Quartz has good crystallinity, evidently apparent by the diffraction peaks at 4.25; 3.34; 2.45; 2.28; 2.12; 1.82 and 1.54 Å. Chlorite and kaolinite group minerals—at peaks of 7.12; 3.53; 14.2; 4.68, 2.33 Å, etc. It should be noted that all minerals are in close proximity to one another and appear to be joined by a mass of siliceous and ferrous cementing mass. Siltstones typically have a water absorption of 1–2% and a porosity of 3–5%. Compressive strength varies on average from 10 to 40 MPa.

The complete absence, or small amount, of clay minerals is the main difference between screenings and clays. Because of this, if the screenings contain argillite-like clays or mudstones they are of low plasticity or completely devoid of this property, even when being crushed. Despite the fact that fired samples based on screenings sinter well and with

a rather high strength index, it is impossible to form products using extrusion technology. As a result, we noticed their poor formability. The firing temperature and the degree of grinding of screenings are the main factors that affect the characteristics of fired samples.

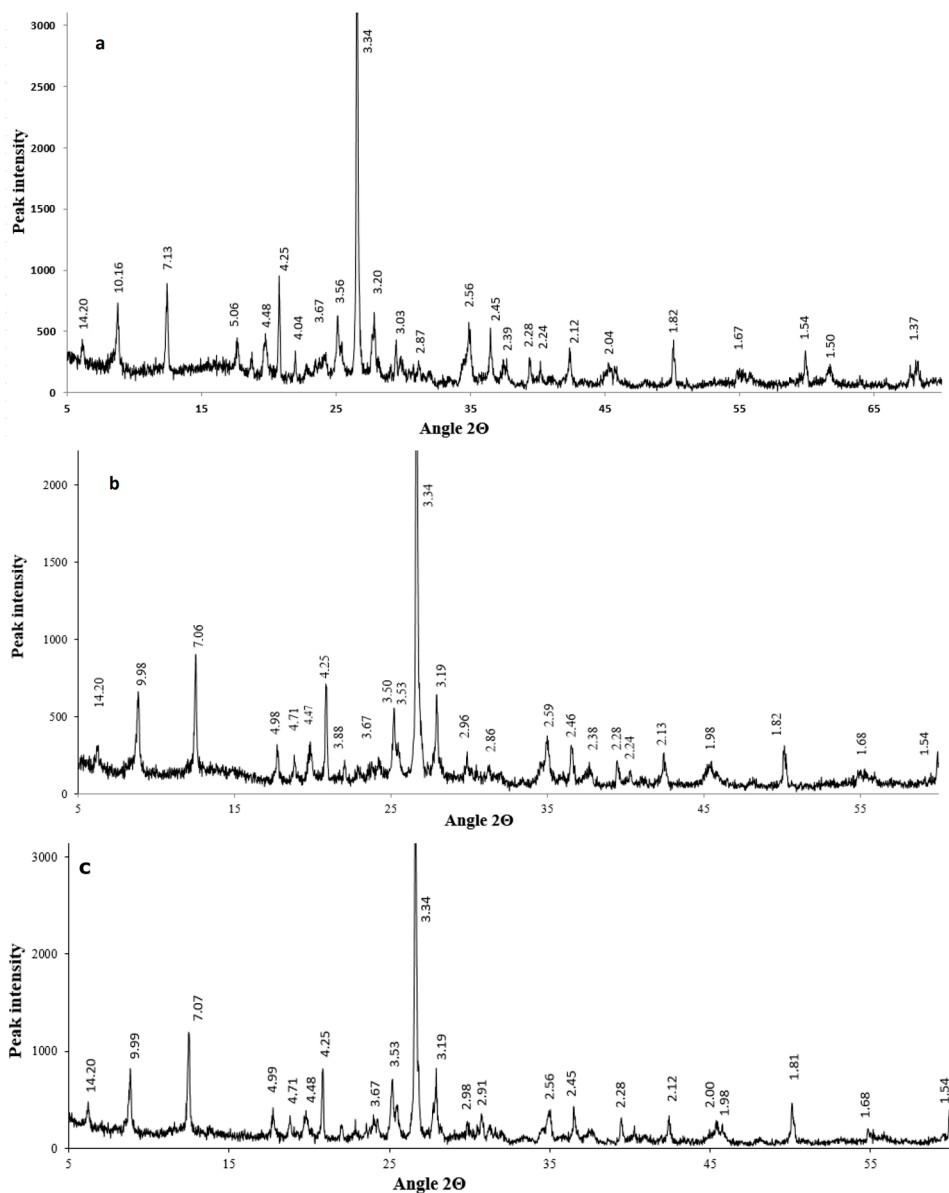

**Figure 2.** X-ray diagrams of common screenings of waste heap processing of different companies of Eastern Donbass. Rostov region, Russia: (**a**) Coal technologies LLC, (**b**) Sulinugol LLC, (**c**) Piramida LLC.

In order to optimize the composition of the raw charge, plastic clays were introduced into the mixture. Siliceous clays were chosen as a plasticizing additive. They have medium plasticity and medium dispersibility properties (47–53% filling below 1 micron) and are commonly used in the southern part of Russia.

Amongst significant tasks are the reduction of product costs and the cost of firing, which can reach up to 25%. The above prerequisites confirm that the introduction of finely dispersed products of waste heap processing into the composition of raw material mixtures can be called predominantly optimal. Initially, they appear in the form of a pulp from the technological cycle and dehydrate naturally later, which is why in most cases they are called coal sludge. The coal content in them is usually 30–50%. Most often, the mineral part is represented by mudstones, because they are moderately plastic. This property is

present due to low strength and transitions to fine fractions during technological processing (crushing, screening, moisturizing). In our case, coal sludge performs three main functions in ceramic masses, acting as additives:

- Pore-forming, reducing the thermal conductivity and density of products;
- Plasticizing, known as "technological lubricant" because the external friction created during molding items is greatly reduced by finely dispersed coal particles;
- Fuel-like, leading to a significant reduction in the use of external fuel for firing (gas), or its complete reduction in use.

It should be emphasized that the cost of a calorie of heat from coal sludge, when compared with gas or pure coal, is 15–20 times lower. However, it should be taken into account that for firing wall products the coal content should not exceed 8–10% in the composition of the ceramic mass, otherwise more heat will be released than is necessary during firing [15–22].

Standard methods for testing samples were used in accordance with Russian GOSTs in order to provide a comprehensive study of the physical and mechanical properties and structure of ceramic materials based on the products of waste heap processing.

The chemical compositions of PPT and fired samples were determined in accordance with the requirements of GOST 21216-2014 "Clay raw materials. Methods of testing".

Laboratory and technological samples of PPT weighing 50–100 kg were taken in specific areas of their accumulation, storage, and processing. They were then formed from samples that did not contain plant, soil, or foreign inclusions. Basic samples of raw materials were preliminarily dried to an air-dry state if necessary, were crushed in a laboratory jaw crusher, and fractionated according to grain composition.

Using the compression test method, after grinding the raw material was moistened to the required molding moisture content and aged under conditions excluding drying of the mass for 24–48 h. Next, molding of standard samples was carried out: cubes with dimensions of $50 \times 50 \times 50$ mm, bricks with dimensions of $67 \times 30 \times 15$ mm and beams with dimensions of $135 \times 30 \times 15$ mm and $160 \times 40 \times 40$ mm. For control tests, brick samples of standard sizes with dimensions of $250 \times 120 \times 65$ mm and $200 \times 100 \times 50$ mm were made. The pressing pressure ranged from 10 to 50 MPa. The density in terms of the solid phase was determined by the calculation and control of dry samples.

Freshly molded samples were kept under natural conditions for a day in the absence of drafts, and then dried in a desiccator for 24 h at a maximum temperature of $90 \pm 5$ °C. After drying, the samples were examined with fixation of all changes in appearance, and air shrinkage was determined as well. Firing was carried out in laboratory electric furnaces with a chamber volume of 50–200 L, with automatic control of the firing mode with an average temperature rise rate of 1–3 °C/min, and exposure at a maximum temperature of 900–1150 °C for 1.5–2 h.

The reliability of the results obtained in the course of research was ensured by control using statistical methods. The data obtained obey the law of normal distribution of random errors.

When processing the obtained results, the confidence interval (with a confidence probability equal to 0.95) and the standard deviation were determined.

## 3. Results

The results of our previous studies [18–21] showed that the most appropriate materials for production of large-sized ceramic stones were medium-grained materials of waste heap processing and fine-grained materials—coal sludge. The first are the basic raw materials. Their secondary grinding to a fraction of less than 0.3–0.6 mm is the main technological operation. It is the degree of their grinding that largely determines the strength of the samples after firing. Figures 3 and 4 show how the degree of screen grinding and the firing temperature affect the compressive strength and water absorption of fired samples. It can be seen from the graphs that a fraction size of 0–0.315 and 0–0.63 mm guarantees the

strength characteristics of products, taking into account their porosity up to 50% at firing temperatures of 950–1050 °C [22–30].

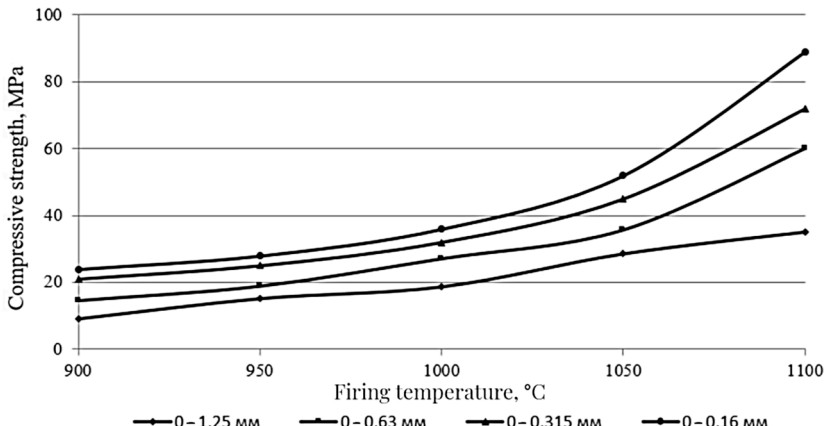

**Figure 3.** Dependence of the compressive strength of samples on firing temperature and degree of screening grinding.

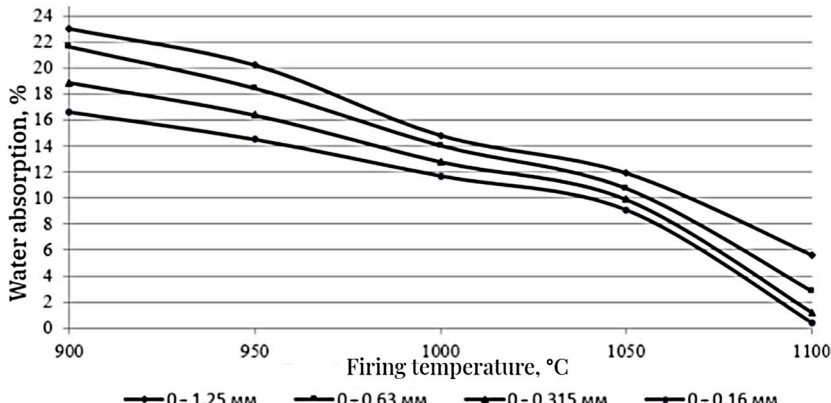

**Figure 4.** Dependence of water absorption of samples on firing temperature and degree of screening grinding.

Based on our previously conducted tests on the selection of the optimal composition of ceramic mass, it was established to take 60% of waste heap processing screenings, 25% of siliceous clay, and 15% of coal sludge. At the same time, the most suitable degree of screening grinding appeared to be from 0 to 0.315 mm and firing temperature—1000 °C.

Fine-grained materials obtained in the process of waste heap processing are a combined additive. They provide the necessary plasticity and cohesion of the raw mixture already at a content of 15–20%, and they are also a fuel-burning additive. This allows for a reduction in the density of the ceramic shard and a large decrease in the amount of gas required for firing.

However, the amount of coal sludge should not exceed 22% when the content of coal in the sludge is around 35%, so the amount of fuel introduced with the raw materials should not exceed 80%. At these ratios, the mass's plasticity only reaches 7–8 units, which is insufficient for the extrusion molding of products. For this reason, coal sludge has already been chosen as a combination additive. It also serves as a fuel-burning additive, which can reduce the density of ceramic shards and significantly reduce gas consumption for firing. Its cost per calorie of released heat is 15–20 times lower (200–400 rubles per ton) than that of gas and clean coal (7000–9000 rubles per ton).

Coal particle combustion during firing contributes to the formation of a porous structure of the ceramic material, which significantly reduces the density and thermal conductivity of products. However, the strength of the products decreases at the same time.

Therefore, determining the optimal amount of introduced sludge is quite a complicated technological task. The dependence of compressive strength of samples with 22% of coal sludge on firing temperatures of screenings with fractions of 0–0.63 and 0–0.315 mm is represented on Figure 5.

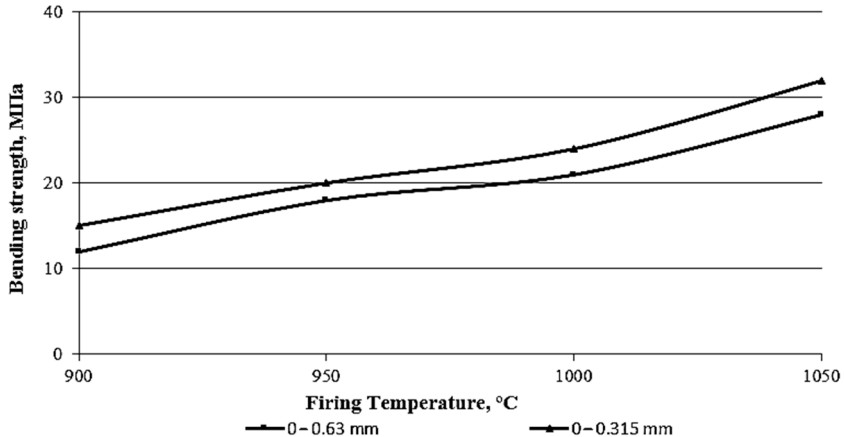

**Figure 5.** The dependence of compressive strength of samples with 22% of coal sludge on firing temperatures of screenings with fractions of 0–0.63 and 0–0.315 mm.

It can be seen that at 22% of coal sludge content in the raw mixture, the strength of samples is great enough to obtain highly effective large-sized ceramic blocks with 50% porosity and a strength grade of at least M100. The firing temperature should be between 950–1050 °C, depending on the desired strength. It should be noted that given the strength of the ceramic material itself, it is possible to improve the porosity of goods. Currently produced large-sized ceramic blocks with vertically arranged voids obtain porosity up to 60%, with the horizontally arranged voids up to 70%, however, their strength indicators are significantly lower.

Fuel can comprise up to 80% of the raw mixture composition. In terms of products, this means that up to 8–10% of pure coal can be contained in the dry raw material. Considering that, on average, the coal content in coal sludge is about 30%, then, accordingly, their content in the raw mixture can be up to 25–30%. The burning of coal particles during firing contributes to the formation of a finely porous structure of the shard, which can significantly reduce the thermal conductivity of products. However, along with this, the introduction of sludge contributes to a decrease in the strength of products and it is quite a difficult task to determine the optimal sludge content, taking into account all the necessary technological parameters. The density of the shard is 1500–1600 kg/m$^3$. Taking into account the stone porosity, the average density of products will be 750–800 kg/m$^3$. At the same time, there is a margin of safety of the shard to increase the porosity and reduce the density of products. Currently produced ceramic blocks with a vertical arrangement of voids have a porosity of about 60%, with a horizontal arrangement of voids up to 70%. However, the latter can only be used for building enclosed structures.

Our research in this area has contributed to the development of a concept that is especially pertinent for coal-mining and residing regions: the creation of large-sized porous ceramic stones using stiff extrusion technology and waste heap screens.

Stiff extrusion technology is the leader in most countries and is now regarded as the most progressive in the manufacture of ceramic wall goods. It appeared due to the introduction of powerful extruders that can mold raw products at a reduced humidity and under pressure of more than 2.5 MPa in the press head. Due to this, the molded raw products are endowed with strength, which contributes to their laying on kiln trolleys without being fully dry, which significantly reduces the cost of drying and delivery of raw material. If an operation such as transferring raw material from drying to firing trolleys is ignored, then it will be possible to significantly reduce technological costs, the number of

machines, and to simplify the technological scheme. Capitalization for the construction of the newest brick shops and factories using the stiff extrusion technology is 30–35% less than when building shops using the ordinary plastic extrusion method, and less industrial space is also needed.

The technology for the production of large-format ceramic blocks from waste heap processing without auxiliary transportation equipment, using the technology of stiff extrusion for molding products, is shown in Figure 6.

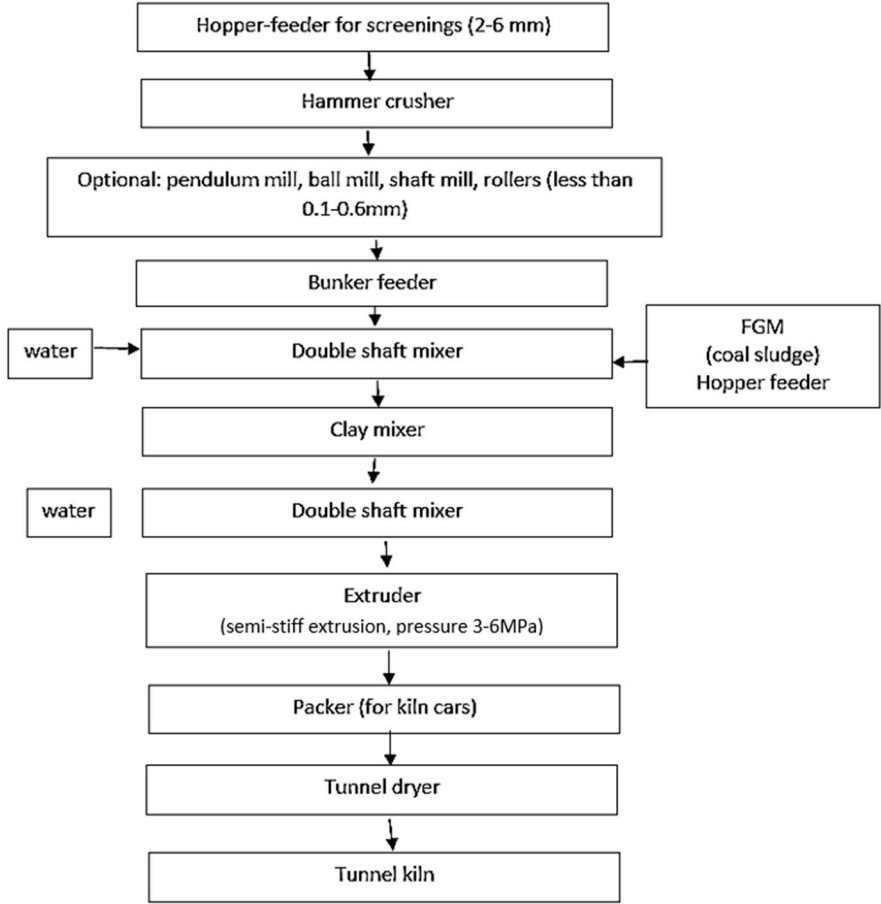

**Figure 6.** Technological scheme for the production of high-performance ceramic blocks based on products of waste heap processing.

## 4. Discussion

Distinctive features of our research and proposals from those available in this area [13, 22–37] are described in this section.

The main procedure in the preparation of the raw mixture is the grinding of screenings to a fractional composition ranging from 0.16 to 0–0.63 mm. It is most rational to use a hammer mill for primary grinding, for secondary—depending on the petrographic composition of the screenings and the required grain composition—one of the following can be chosen: a pendulum mill, a shaft mill, a ball mill, or rollers. Ultimately, everything will be determined by the initial operating costs. Fine-grained processing materials do not require grinding because they are finely dispersed in the initial state. The task is to thoroughly mix the mixture's components while carefully controlling its moisture content, which for the given raw masses and the molding method is 14–15%. Therefore, two mixers and a clay grinder are provided in the production line. Products are formed on extruder presses. The presence of a carbon component is especially beneficial for stiff extrusion, as the carbon is a kind of "lubricant" during molding. The main task is to select the technological parameters of molding (pressure in the press head, vacuum depth, steam

temperature, molding temperature, beam output speed, etc.) to obtain defect-free products with strength sufficient for mounting on kiln trolleys. Robots should be used for product placement, since it is only with their help that it is possible to guarantee the laying of blocks without the risk of damage and deformation.

The absence of stacking products on drying trolleys and removing them from drying trolleys as a separate operation greatly simplifies the technological scheme of production. Drying as a separate operation may be eliminated for these raw masses with a sufficient length of the furnace (150 m or more). The main task during firing is to make sure that the coal component is completely burned out. According to the scheme, firing can be carried out entirely on coal in this case: 80% of the fuel in the products themselves, 20% of external fuel, and thus external combustion. It should be said that the presence of fuel inside the products contributes to uniform firing. The introduction of sawdust in the amount of 1–2% improves the firing conditions by increasing the combustion zone, since sawdust ignites at much lower temperatures than coal and has more volatile components in its composition. Waste heat of the furnace can be used for other needs of the enterprise.

The products which are ready to be fired are laid according to a certain scheme, fully corresponding to the type of fuel used and the design of the furnace.

In order to load and unload the stone, a special rack must be stable enough and convenient to use at the same time. Its design is important in the formation of heat exchange processes and heat distribution between products. The type of fuel affects the type of special rack [31,32].

The following are methods for laying high-performance ceramic stones of the 14.3 NF format on drying and kiln trolleys (Figures 7 and 8).

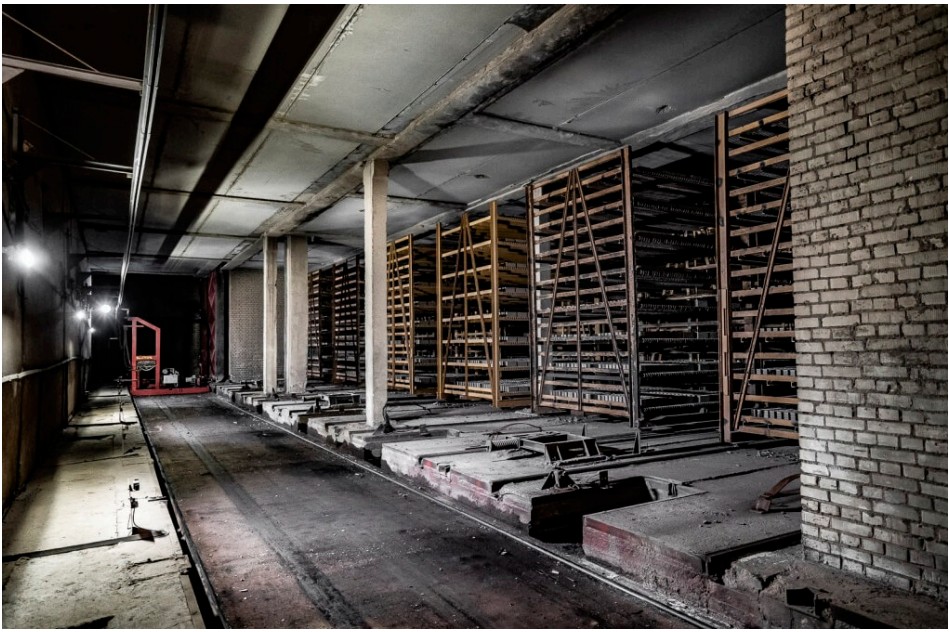

**Figure 7.** Stacking products on drying trolleys.

A rational firing mode is one in which the products are fired without defects, and in accordance with technical indicators in the shortest possible time, with minimal fuel consumption.

The totality of the results of laboratory and technical tests of raw materials and the results of experimental firing of products or their models, under conditions similar to those in real equipment for which this mode is set, is a predominantly advantageous and often used method for determining a rational mode.

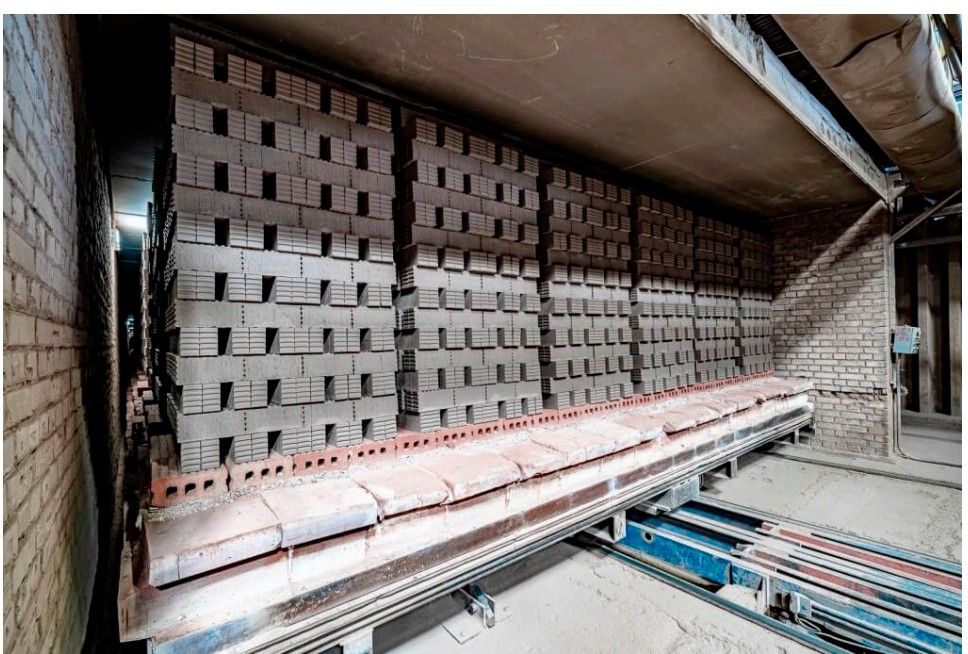

**Figure 8.** Stacking products on kiln trolleys.

The firing cycle includes a period of heating, holding in the region of maximum temperatures, and cooling for ceramic products.

The firing temperature is the most important condition affecting the degree and speed of sintering. During solid-phase sintering in the process of firing wall ceramic products, an increase in the firing temperature accelerates the diffusion of the substance and, accordingly, the sintering process itself. An increase in temperature helps the system to achieve the conditions for liquid sintering, improving the wetting ability of the liquid phase and decreasing the viscosity [13,35,36].

We identified several key stages depending on the firing temperature in ceramic masses based on screenings:

1. Getting rid of free water. There is residual moisture in the products in the range of 2–4% after drying. It is removed in the temperature range up to 200 °C. Initially, the elimination of water occurs from larger pores, then from smaller ones, so this process occurs gradually;
2. Oxidation (burnout) of organic impurities contained in screenings, which begins at temperatures of 250–350 °C. This process is completed at 700–800 °C;
3. Clay dehydration begins at 400–600 °C—removal of chemically bound water;
4. Isolation of iron oxides;
5. At 650–700 °C anthracite ignites, heat is released, and the supply of external heat is interrupted;
6. The formation of a liquid phase is accompanied by sintering from approximately 950 °C. The liquid phase is very important in the sintering of a shard as it helps separate mineral particles to stick together into a single one [37].

The presence of iron content in raw materials with different firing environments leads to different reactions. Thus, the firing of clay in an oxidizing environment leads all the iron into its oxide form, and in the reducing one—into its reduced oxide form, coloring the products in a blue-greenish color. With an increase in the amount of iron in the composition of clay, it darkens more and more and may turn black after firing. In order to improve the quality of products redox firing has been widely used.

Water vapor is supplied to the installed burners at the end of the burning zone. Along with this, the air consumption coefficient decreases to 1. In the next half of the firing zone, there should be a section of the reducing environment covering more than four positions

of the kiln. The oxygen content within this section should not be more than 0.3%, when the largest accumulation of reducing gases (CO + H$_2$) should range from 1.5% to 4%. It is necessary to limit the volume of secondary air entering the second half, along with a decrease in the supply of primary air to 5–6 pairs of the last burners: the firing zones of the kiln in the form of suction and directed through the kiln channel from the cooling zone to the firing zone. The supply of steam to the burners located at the end of the firing zone has an effect on reducing the temperature of the combustion flame when the burners operate with an air flow rate close to 1, thereby eliminating overheating of products at the points of action of the combustion flame.

The aerodynamic mode of the furnace guarantees an overpressure in the second half of the firing zone over the entire cross section of the special cage of about 3 Pa and supports a decrease in the content of secondary air entering the firing zone. This requires the competent use of sand gates, constant monitoring of the trolley fleet, the closest attention to the joints and aprons of the trolleys, as well as special control of the sealing of the firing channel in the mentioned area [38–41].

## 5. Conclusions

The conducted studies of waste heap processing screenings made it possible to identify the following significant conclusions for the production technology of large-sized ceramic blocks:

- Screenings of waste heap processing are raw materials for low-temperature sintering. Water absorption is less than 5% at firing temperatures up to 1100 °C;
- The grind size determines which group raw materials will be assigned to during firing;
- Screenings have a small sintering interval—no more than 50 °C, which is possible only when preparing screenings due to an increase in the content of fine fractions. Moreover, it is the degree of grinding of screenings that makes it possible to regulate the sintering process;
- In the manufacture of stones based on screenings and ordinary bricks, it is necessary to focus on temperatures of 1000—1050 °C;
- When choosing corrective additives for screenings, it is worth taking complex ones as a basis. They increase the plasticity and cohesion of the molding compositions and extend the sintering range. We decided to focus on siliceous clays and coal sludge in the form of such additives.

The key factors influencing the formation of the structure of a ceramic shard based on screenings are the degree of their grinding and the firing temperature. These data are confirmed by numerous laboratory tests.

Other factors also need to be taken into account, such as:

- Stone format, dimensions, porosity;
- Special rack shape;
- Type of firing unit;
- Firing mode;
- Firing environment.

As for the formats, sizes and porosity of ceramic stones, there are some rules: the shape of a ceramic block is a parallelepiped with different lengths of facets, which depends on the type of block. The porosity of ceramic stones can vary from 48 to 55%.

As a result, the conducted research revealed the potential of manufacturing large-sized ceramic blocks from waste heap screenings processing. At the same time, it is possible to reduce the cost of products and achieve an increase in technical and economic indicators for the construction industry as a whole. A technological regulation for the manufacturing process with an updated list of equipment is required for further development.

**Author Contributions:** Conceptualization, K.Y. and E.G.; methodology, K.Y. and E.G.; software, E.G.; validation, K.Y.; formal analysis, K.Y.; investigation, K.Y. and E.G.; resources, K.Y. and E.G.; data curation, K.Y.; writing—original draft preparation, K.Y. and E.G.; writing—review and editing, K.Y.;

visualization, K.Y. and E.G.; supervision, K.Y.; project administration, K.Y.; funding acquisition, K.Y. and E.G. All authors have read and agreed to the published version of the manuscript.

**Funding:** This research received no external funding.

**Data Availability Statement:** The data presented in this study are available on request from the corresponding author.

**Conflicts of Interest:** The authors declare no conflict of interest.

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
