# Peer review of "Production of Large-Sized Ceramic Stones Based on Screenings from Waste Heap Processing Using the Technology of Stiff Extrusion for Molding Products"

_buildings, doi:10.3390/buildings13040845_

Round 1

Reviewer 1 Report

This is a technical report rather than a scientific paper. However, the manuscript may be of interest to the scientific community of construction sciences.

Prior to its acceptance for publication, I recommend the following modifications:

1) The authors must introduce at the end of the introduction section the objectives of their work and the novelty that it brings over other works referenced in the literature.

2) The methodological section should be further improved. The authors must describe all the analytical protocols, patterns used, detection limits of the techniques, replicates performed, statistics,...

3) Compare the results obtained with other similar or different ones obtained by other authors in the literature.

Reviewer 2 Report

The paper deals with an interested topic. However, the manuscript lacks of clarity, and then the authors should improve the quality of the manuscript in order to make it suitable for publication.

In the first 10 rows of the introduction there are many statements without any bibliography, please provide some references.  

Materials and methods section has to be modified, since it is not clear what the authors actually made, and what was taken from the literature. For example there is just the composition on term of oxides, but the authors discussed about grain size, minerals and so on, but there are not any results reported. I recommend to specify all measurements made and describe how did they make it (compressive strength etc-). Moreover, all the “recipes” used should be reported in a table.

Rows 135-136 “The results show that…” the results are described after this statement, so I kindly suggest to move this sentence after showing the results.

Results are quite unclear, for example there is not reported any results about the effect of coal on the structure of the block although the authors discuss it.

Reviewer 3 Report

The manuscript “Production of large-sized ceramic stones based on screenings from waste heap processing using the technology of rigid extrusion for molding products” is more like a technical report rather than academic paper.

 Comments

1.     “1. Introduction” part, the introduction is weak, the research progress for large-sized ceramics stones is missing. The screenings reusing is also not well presented.

2.     “2. Materials and Methods” part, it provides lots of additional information which can be moved to other parts. And the fabrication process is not very clear for other researches to reproduce. Fig. 4 can be moved to this part.

3.     In the result part, it seems showed the mechanical strength and water absorption. The microstructure and phase evolution are missing. It is almost no discussion, it simply showed the mechanical strength and water absorption data. While in the “discussion” part, authors provide some “products” figures. The structure is hard to understand, it is not following a normal form for an academic paper.

Round 2

Reviewer 1 Report

The authors have made a great effort to improve this manuscript. They have responded to all reviewers' considerations, comments, and criticisms. Congratulations.

Reviewer 2 Report

The authors addressed all my comments

Reviewer 3 Report

accept as it is.